# Association between shift work in early pregnancy, snacking, and inappropriate weight gain during pregnancy: The Japan Environment and Children's Study

Satomi Funaki-Ishizu[1]*, Toshio Masumoto[1], Hiroki Amano[1], Shinji Otani[2], Youichi Kurozawa[1], the JECS Group[¶]

1 Division of Health Administration and Promotion, Department of Social Medicine, Faculty of Medicine, Tottori University, Yonago City, Tottori Prefecture, Japan, 2 International Platform for Dryland Research and Education, Tottori University, Tottori City, Tottori Prefecture, Japan

☯ These authors contributed equally to this work.
¶ Membership of the JECS group is provided in the Acknowledgments.
* d20m1016c@tottori-u.ac.jp

**Data Availability Statement:** Data are unsuitable for public deposition because of the ethical restrictions and legal framework of Japan. It is

## Abstract

### Objectives

This study aimed to investigate the relationship between night shifts and snacking behaviors during pregnancy using Japanese maternal-infant longitudinal data from a large-scale study.

### Methods

This study used the Japan Environment and Children's Study dataset jecs-ta-20190930, released in October 2019. After simple analysis using analysis of variance and multiple comparisons, crude odds ratios (cOR) and adjusted odds ratios (aORs) were calculated. To evaluate eating habits, we examined habitual fast food and snacks (e.g., potato chips, corn puffs and tortilla chips) consumption, midnight snacks, and regular missing breakfast.

### Results

There was no significant association between inappropriate weight gain during pregnancy and night shift work in early pregnancy. The aOR for consuming snacks more than once a week after noticing pregnancy for shift workers was 1.34 (95% confidence interval 1.27–1.41; $p < 0.001$) compared with worker without night shiftwork. The aOR for shift workers consuming fast food more than three times a week during pregnancy was 1.40 (95% confidence interval 0.79–2.33; $p > 0.05$).

### Conclusions

Pregnant night shift workers did not show excessive weight gain but had an increased frequency of consumption of snack foods compared with pregnant dayshift workers.

prohibited by the Act on the Protection of Personal Information (Act No. 57 of 30 May 2003, amendment on 9 September 2015) to publicly deposit the data containing personal information. Ethical Guidelines for Medical and Health Research Involving Human Subjects enforced by the Japan Ministry of Education, Culture, Sports, Science and Technology and the Ministry of Health, Labour and Welfare also restricts the open sharing of epidemiological data. All inquiries about access to data should be sent to: jecs-en@nies.go.jp. The person responsible for handling enquiries sent to this e-mail address is Dr Shoji F. Nakayama, JECS Programme Office, National Institute for Environmental Studies.

**Funding:** This research was supported by the Ministry of Environment, Japan, in the form of a grant to the Japan Environment and Children's Study.

**Competing interests:** The authors have declared that no competing interests exist.

## Introduction

In 2012, Kubo et al. estimated that 21.8% (12 million) of Japan's working population was engaged in late-night work, and 11.6% (6.38 million) were engaged in shift work, with a consistently increasing trend [1]. Previous studies by Bøggild et al. and Ha et al. indicated that shift work is a risk factor for cardiovascular disease, increased body mass index (BMI), overweight, and glucose intolerance [2, 3]. Shift workers tend to exhibit an increased risk for diabetes compared with those who only work day shifts, and a dose-response relationship with the average annual number of night shifts has been observed [4–7].

However, shift work can contribute to nutritional deficiencies and may be a risk factor for obesity in workers [8–12]. By Hemiö et al. study of 1,478 airline workers, shift workers had unhealthy nutritional habits, including lower consumption of fruits and vegetables and significantly higher energy intake of saturated fats, especially among female shift workers, compared with day shift workers [8]. In a Samhat et al. study of shift work and eating habits among nurses in Lebanon, 78.2% of nurses working night shifts had meals at different times than usual, resulting in less food consumed during the day and more snacks consumed at night [9]. The most commonly consumed snacks were soft drinks and potato chips. In previous studies, Heath et al. and Hulsegge et al. indicated that sweets and low-quality snacks increased significantly during sleep restriction and night shifts [10, 11]. These findings suggest that unhealthy snacking behavior and increased consumption of sweets may play a role in obesity among shift workers. As a result of poor nutrition, shift work has been suggested to contribute to increased BMI and is, therefore, a possible risk factor for obesity [12].

Working conditions and work environments for pregnancy/childbirth have improved in recent years. However, research on weight gain among pregnant women who engage in shift work is scarce because pregnant women have typically been excluded from studies on this topic [5]. Previous studies have shown that shift work pregnant women undertake can negatively affect childbirth and infants [13–17]. Still, as far as we have been able to research, very little past research exists on the association between shift work and unhealthy snacking during pregnancy.

Thus we aimed to investigate the effects of working night shifts on excessive weight gain and snacking behavior during pregnancy using Japanese longitudinal data for mothers and children in the current study.

## Materials and methods

### Study design and data sources

The Japan Environment and Children's Study (JECS) is a nationwide, multi-center, prospective birth cohort study of approximately 100,000 fetuses in Japan conducted by the Ministry of the Environment. The JECS involves 15 regional centers in Japan: Hokkaido, Miyagi, Fukushima, Chiba, Kanagawa, Koshin (Yamanashi and Nagano), Toyama, Aichi, Kyoto, Osaka, Hyogo, Tottori, Kochi, Fukuoka, and Southern Kyushu/Okinawa (Kumamoto, Miyazaki, Okinawa) [18]. The recruitment period for JECS was from January 2011 to March 2014. JECS launched Follow-up with a plan to continue until children reach 13. However, with the revision of the basic strategy in 2022, continuing the survey after children reached the age of 13, and the overall survey period was set until all participants (children) came approximately 40 years of age (around 2054) [19]. Each regional center requested cooperation from all obstetric facilities where pregnant women residing in the study area were likely to receive medical examinations and deliver babies and designated all obstetric facilities that agreed to cooperate as cooperating medical institutions. In addition, with the cooperation of the local governments

concerned, this survey was introduced to all pregnant women residing in the study area at the time of issuance of the Maternal and Child Health Handbook at the local government's Maternal and Child Health Handbook issuance counter (the Mother-Child Health Handbook is an official booklet which all expecting mothers in Japan are given complimentary when they become pregnant to receive municipal services for pregnancy, delivery, and childcare) [18]. JECS members targeted the recruitment rate at more than 50% of all eligible mothers.

The present study established a participation request-target of 50% or more of the total number of births (Vital Statistics) of children born to participants (mothers) in the study area during the survey period. Follow-up surveys were conducted in early pregnancy (at the time of enrollment), mid and late pregnancy, delivery, one month after birth, and every six months after the infant was six months old; these follow-ups involved questionnaires, interviews, and collection of various biological materials, as needed [18]. For the survey data obtained in this study, we used the dataset jecs-ta-20190930, routinely cleaned by JECS at three years of age for participating children.

### Selection protocol for research participants (Fig 1)

The present research used the JECS dataset jecs-ta-20190930, released in October 2019 [18]. In this study, the number of enrolled mothers was 104,062. We excluded mothers with multiple-child births from the study (n = 1,992) because of differences in definitions of adequate weight gain during pregnancy compared with mothers of singletons. Additionally, we excluded participants with second and subsequent pregnancies because of the potential impact on maternal eating habits and weight gain during pregnancy (n = 5,605). The present study includes only production; thus, in this study and cases of miscarriages (< 34 w) and stillbirths were excluded (n = 3,520). In addition, pregnant women with pregnancy complications affecting the course of pregnancy, such as severe gestational hypertension, gestational diabetes, placenta previa, amniotic fluid deprivation, amniotic fluid overload, and placental abruption, were excluded (n = 6,485). This research excluded the 3,536 people for whom information on employment status was. Finally, we investigated 82,924 mothers in the analysis in this study.

### Exposure (Fig 2)

**Shift work.** We defined nonworkers and workers based on questions about their working conditions in early pregnancy. For this study, we defined shift workers as those who responded to the question "Number of night shifts per month" with one or more. The questionnaire included a supplemental question, "How many days per month do you work during hours other than day shift hours (approximately morning to evening)?" Based on the question, we defined workers who indicated they completed one or more night shifts per month as shift workers. Exposure factor categories included nonworkers, day-shift-only workers, and shift workers. The reference group was workers that only worked day shifts.

### Outcomes

**Inappropriate weight gain during pregnancy.** The non-pregnant weight obtained from the physician took the questionnaire as each woman's pre-pregnancy weight. JECS calculated weight gain during pregnancy from the weight immediately before delivery, obtained from medical record transcripts. This study defined the "appropriate weight gain" range during pregnancy based on the Optimal Weight Gain Chart for Pregnancy published by the Ministry of Health, Labour and Welfare in 2006 [20]. This chart classified appropriate weight gain as 9–12 kg for underweight women (BMI: < 18.5 kg/m2), 7–12 kg for normal-weight women (BMI: 18.5–24.9 kg/m2), and within 5 kg for women with obesity (BMI ≥ 25 kg/m2). In the

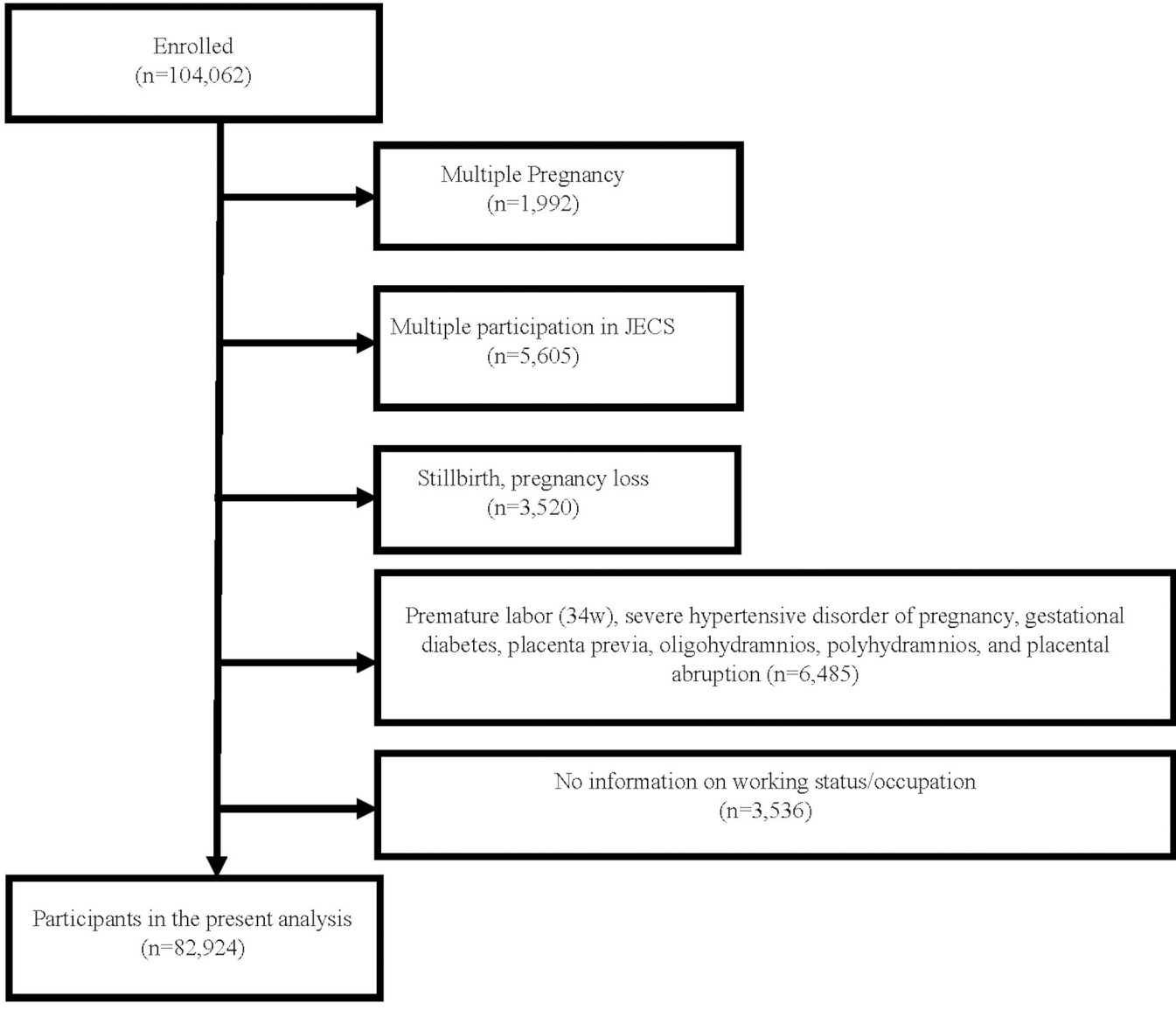

**Fig 1. Flow diagram of selection participants of the present study.**

present study, we defined individuals outside this range as either "less weight gain" or "excess weight gain" and classified as "inappropriate weight gain".

**Dietary habits.** We examined unhealthy eating habits, including the frequency of eating fast food (e.g., French fries, pizza, donuts) for breakfast, lunch, or dinner; the frequency of missing breakfast; frequency of eating at night; and the frequency of consuming snacks (e.g., potato chips, corn puffs, and tortilla chips) [21]. These foods were related to items found to be affected by night shifts in previous studies [9].

We used data from responses to a questionnaire survey administered in mid/late pregnancy for fast food, first trimester and mid/late pregnancy for the other factors.

**Frequency of eating fast food.** We defined "habitual fast food consumers" as individuals who indicated that they ate fast food three or more times a week within the last month when

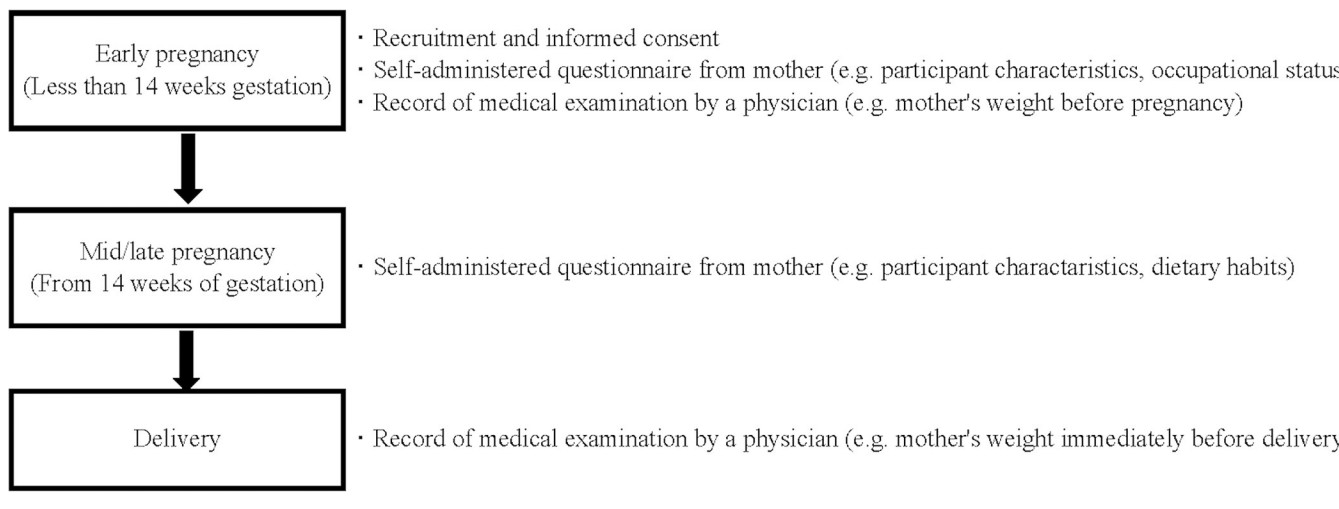

**Fig 2. The flow of study participants in JECS from enrollment to delivery.**

answering the question in mid/late pregnancy. Those who ate fast-food less than three times a week were defined as "non-habitual fast food consumers".

We defined "chronic fast food consumers" as individuals who indicated that they ate fast food three or more times a week within the previous month when answering the question in mid/late pregnancy. We defined those who ate fast-food less than three times a week as "non-habitual fast food consumers".

**Frequency of skipping breakfast.** We defined those who said they skipped breakfast more than thrice a week as habitually missing it.

**Frequency of having midnight snacks.** We defined those who reported they had midnight snacks three or more times a week as chronic midnight snackers.

**Frequency of consuming snack foods.** The questionnaire asked about the frequency of eating snack foods and the amount of them at a time. In the questionnaire, potato chips were a typical snack food example. Snack foods included snacks besides potato chips (e.g., corn puffs and tortilla chips). Those who reported consuming snack foods at least once a week after gestation were considered regular snack consumers.

## Statistical analyses

Study participant characteristics were described using mean and standard deviation (SD) for continuous variables and number and percentage (%) for categorical variables. Proportions were compared for inappropriate weight gain during pregnancy and poor dietary habits using the chi-square test, analysis of variance (ANOVA) and Bonferroni's multiple comparisons to examine which groups had significant differences. After a simple analysis of variance and multiple comparisons, we calculated crude odds ratios (cORs) for items significantly different in the chi-square tests and multiple comparisons. By logistic analysis, we estimated ORs and 95% confidence intervals (CIs) for the relationship between shift work in early pregnancy and weight gain or poor dietary habits during pregnancy. We adjusted these estimates for maternal age and other covariates. The covariates considered in this study were: maternal age at gestation, maternal education, partner's education, presence of partner, presence of children at the time of pregnancy, annual income, existence of obesity before pregnancy, major occupational category at awareness of pregnancy, whether or not the participant was a medical professional (e.g., medical doctors, dentists, pharmacists, public health nurses, midwives, nurses, medical

technicians, and other health workers), 6-item Kessler Psychological Distress Scale score in early and mid/late pregnancy (cut-off: 9 points) [22], presence of maternal social support (company of friends/neighbors to whom you can talk casually about the participant's concern), presence of fertility treatment, history of miscarriage, history of alcohol consumption, history of smoking, presence of exercise habits (vigorous physical activity, moderate physical activity, and walking for > 10 minutes consecutively on the short version of the International Physical Activity Questionnaire) in mid/late pregnancy [23, 24], history of morning sickness during the first 12 weeks of pregnancy, and whether the participant was aware of the need to avoid being overweight during pregnancy. We calculated adjusted ORs (aORs) for items significantly different in the cORs. We considered 2-sided $p < 0.05$ to be statistically significant. We performed all analyses with R statistical software version 4.3.1 (R Foundation for Statistical Computing) [25].

## Sensitivity analyses

We examined whether there would be a significant discrepancy in the results if we changed the definitions as follows:

1. We altered the definition of night shift from once more a month to at least once a week.

2. We changed the cut-off value for the frequency of snack foods intake from once to three times a week.

3. We altered the cut-off value for the frequency of fast food intake from three to once per week or five times per week.

We also used the multiple substitution method to determine if there were any inconsistencies or significant changes in the results when missing values were supplemented in the simulation using mice, miceadds, and norm2 packages [26–28].

**Ethical approval.**    The JECS protocol was reviewed and approved by the Ministry of the Environment's Institutional Review Board on Epidemiological Studies (no. 100910001) and the Ethics Committees of all participating institutions. Written informed consent was obtained from all study participants.

## Results

Of the 82,924 participants analyzed, 27,008 (20.5%) were not working at the time they learned of their pregnancy, 49,489 (51.2%) were working without night shifts, and 6,427 (7.75%) were working night shifts (Tables 1–3). The proportion of women who gained too little weight during pregnancy compared to nonworkers was significantly lower among day-shift-only and shift-worker women. However, there was no significant difference between day-shift-only and shift-worker women. The proportion of women who gained too much weight during pregnancy was more significant for shift-only and day-shift-only workers compared to the nonworking group. However, there was no significant difference between shift-only and day-shift-only workers.

The mean daily caloric intake for each group after realizing pregnancy was nonworker: 1744 kcal (SD: 726), dayshift: 1722 kcal (SD: 756), and nightshift: 1767 kcal (SD: 1056), with significant differences ($p<0.005$ by ANOVA) (Table 4). Multiple comparisons (Bonferroni method) for dayshift against nonworker ($p<0.005$) and nightshift against dayshift ($p<0.005$) with superior differences. However, since the difference in mean caloric intake for each group was less than 50 kcal, we considered the difference was not significant enough to make a clear difference in weight gain during pregnancy. Thus in the present study, we did not adjust for caloric intake. According to the "Dietary Guidelines for Pregnant Women" (2006) [29], at the

Table 1. Medical characteristics of the study participants.

| Variables | All (N = 82,924) | Nonworkers (n = 27,008) | Workers without night shifts (n = 49,489) | Workers with night shifts (n = 6,427) | Missing value |
|---|---|---|---|---|---|
| Age (years), mean (SD) | **30.7 (5.01)** | **31.0 (5.10)** | **30.6 (4.98)** | **29.9 (4.76)** | **19** |
| BMI (kg/m2) before pregnancy, mean (SD) | 21.1 (3.17) | 21.2 (3.28) | 21.0 (3.11) | 21.3 (3.23) | 897 |
| Underweight (BMI <18.5 kg/m2) before pregnancy, n (%) | 13,465 (16.2) | 4,495 (16.6) | 8,045 (16.2) [‡] | 925 (14.4) [‡§] | |
| Normal weight (BMI 18.5–24.9 kg/m2) before pregnancy, n | 60,619 (73.1) | 19,348 (71.6) | 36,527 (73.8) [‡] | 4,744 (73.8) [§] | |
| Obesity before pregnancy (BMI ≥25 kg/m2), n | 7,943 (9.6) | 2,823 (10.5) | 4,423 (8.9) [‡] | 697 (10.8) [§] | |
| Mean weight gain during pregnancy, kg (SD) [#] | 10.4 (4.97) | 10.0 (3.94) | 10.7 (5.52) [‡] | 10.5 (4.05) [‡] | 2,562 |
| Pregnant women who gained too little weight during pregnancy, n (%)[#] | 11,810 (14.2) | 4,362 (16.2) | 6,609 (13.4) [‡] | 839 (13.1) [‡] | 2,511 |
| Pregnant women with normal weight gain during pregnancy, n (%)[#] | 39,201(47.3) | 13,008 (48.2) | 23,224 (46.9) [‡] | 2,969 (46.2) [‡] | 2,631 |
| Pregnant women who gained too much weight during pregnancy (%)[#] | 29,282 (35.3) | 8,550 (31.7) | 18,284 (36.9) [‡] | 2,448 (38.1) [‡] | |
| Morning sickness during the first 12 weeks of pregnancy | 67,837 (81.8) | 22,963 (85.0) | 39,647 (80.1) | 5,227 (81.3) | 1,084 |
| Psychological distress in first trimester, n (%) | | | | | 304 |
| K6 < 9 | 73,046 (88.1) | 23,777 (88.0) | 43,667 (88.2) | 5,602 (87.2) | |
| K6 ≥ 9 | 9,574 (11.5) | 3,133 (11.6) | 5,632 (11.4) | 809 (12.6) | |
| Psychological distress in second/third trimesters, n (%) | | | | | 886 |
| K6 < 9 | 73,804 (89.0) | 23,986 (88.8) | 44,169 (89.3) | 5,649 (87.9) | |
| K6 ≥ 9 | 8,234 (9.9) | 2,729 (10.1) | 4,790 (9.7) | 715 (11.1) | |
| Pregnancy loss history, n (%) | | | | | 0 |
| 0 | 66,707 (80.4) | 20,835 (11.1) | 40,449 (81.7) | 5,423 (84.4) | |
| 1 | 12,791 (15.4) | 4,786 (17.7) | 7,195 (14.5) | 810 (12.6) | |
| 2 | 2,673 (3.2) | 1,074 (3.98) | 1,441 (2.91) | 158 (2.46) | |
| 3 | 753 (0.9) | 313 (1.16) | 404 (0.82) | 36 (0.56) | |
| In vitro fertilization and embryo transfer, n(%) | | | | | 0 |
| No | 79,226 (95.5) | 25,764 (95.4) | 47,259 (95.5) | 6,203 (96.5) | |
| Yes | 3,698 (4.5) | 1,244 (4.6) | 2,230 (4.5) | 224 (3.5) | |

BMI, body mass index; SD, standard deviation.

#Proportions were compared for inappropriate and normal weight gain during pregnancy using the chi-square test and Bonferroni's multiple comparisons to examine which groups had significant differences. Mean weight gain during pregnancy using the ANOVA and Bonferroni's multiple comparisons to examine which groups had significant differences.

§ p<0.05 for the group of workers without night shift.

‡ p<0.05 for the nonworker group.

time the study was conducted, the daily caloric intake standards during pregnancy were 1750–2400 kcal for early pregnancy (<16 weeks), 1950–2600 kcal for mid-pregnancy (16–28 weeks), and 2200–2850 kcal in the mid-term (16–28 weeks) and 2200–2850 kcal after 28 weeks of gestation. All groups tended to have slightly lower daily caloric intake than these standards.

The percentages of those who consumed snack foods at least once a week since becoming aware of pregnancy (47.7%) were significantly higher among mothers who worked night shifts in early pregnancy than those of other groups (Table 5). Those who consumed fast food more than three times a week within the last month (at mid/late pregnancy) were slightly more likely to be in the night shift group (workers without shift work 0.8% vs. shift workers 1.2%). In early pregnancy, this was significant for those with night shifts but not for the dayshift-only group.

**Table 2. Socioeconomic status of the present study participants.**

| Variables | All (N = 82,924) | Nonworkers (n = 27,008) | Workers without night shifts (n = 49,489) | Workers with night shifts (n = 6,427) | Missing value |
|---|---|---|---|---|---|
| **Education level, n (%)** | | | | | 1,101 |
| Junior high school or high school | 29,314 (35.4) | 10,416 (38.6) | 17,491 (35.3) | 1,407 (21.9) | |
| Junior college or vocational school | 34,370 (41.4) | 10,726 (39.7) | 19,669 (39.7) | 3,975 (61.8) | |
| University | 16,904 (20.4) | 5,206 (19.3) | 10,790 (21.8) | 908 (14.1) | |
| Graduate school | 1,235 (1.49) | 267 (0.9) | 909 (1.8) | 59 (0.9) | |
| **Partner's education level, n (%)** | | | | | 1,611 |
| Junior high school or high school | 35,475 (42.8) | 11,158 (41.3) | 21,640 (43.7) | 2,677 (41.7) | |
| Junior college or vocational school | 8,353 (22.1) | 5,501(20.4) | 11,009 (22.2) | 1,843 (28.7) | |
| University | 23,733 (28.6) | 8,372 (31.0) | 13,784 (27.9) | 1,577 (24.5) | |
| Graduate school | 3,752 (4.52) | 1,448 (5.4) | 2,098 (4.2) | 206 (3.2) | |
| **Married or cohabiting, n (%)** | 78,929 (95.2) | 26,210 (97.0) | 46,783 (94.5) | 5,936 (92.4) | 347 |
| **Availability of social support, n (%)** | 80,994 (97.7) | 26,321 (97.5) | 48,374 (97.7) | 6,299 (98.0) | 1,094 |
| **Annual household income (million Japanese yen), n (%)** | | | | | 6,461 |
| <2 | 4,223 (5.09) | 1,491 (5.5) | 2,509 (5.1) | 223 (3.5) | |
| 2–3.9 | 26,307 (31.7) | 9,758 (36.1) | 15,096 (30.5) | 1,453 (22.6) | |
| 4–7.9 | 37,559 (45.3) | 11,755 (43.5) | 22,553 (45.6) | 3,251 (50.6) | |
| 8–9.9 | 5,105 (6.16) | 859 (3.2) | 3,505 (7.1) | 741 (11.5) | |
| ≥10 | 3,269 (3.94) | 784 (2.9) | 2126 (4.3) | 359 (5.6) | |

When the reference group was workers without night shift, the cORs for mothers who had consumed snacks at least once a week after becoming aware of pregnancy for those who worked at least one night shift per month in early pregnancy were 1.06 (95% CI 1.03–1.10; p < 0.05) for nonworkers and 1.34 (95% CI 1.27–1.41) for shift workers (Fig 3). The cOR for

**Table 3. The occupational section at the time of awareness of pregnancy.**

| | All (N = 82,924) | Nonworkers (n = 27,008) | Workers without night shifts (n = 49,489) | Workers with night shifts (n = 6,427) | Missing value |
|---|---|---|---|---|---|
| **Occupational section awareness of pregnancy, n (%)** | | | | | 28,490 |
| Managers | 455 (0.6) | 0 | 430 (0.9) | 25 (0.4) | |
| Professionals and technicians | 18,246 (22.0) | 0 | 14,155 (28.6) | 4,091 (63.7) | |
| Clerical support workers | 14,015 (16.9) | 0 | 13,783 (27.9) | 232 (3.61) | |
| Sales workers | 4940 (5.96) | 0 | 4,696 (9.49) | 244 (3.80) | |
| Service workers | 12,874 (15.5) | 0 | 11,371 (23.0) | 1,503 (23.4) | |
| Protective service workers | 198 (0.24) | 0 | 137 (0.28) | 61 (0.95) | |
| Skilled agricultural, forestry, and fisheries workers | 354 (0.43) | 0 | 346 (0.70) | 8 (0.12) | |
| Craft and manufacturing workers | 2796 (3.4) | 0 | 2,632 (5.3) | 164 (2.6) | |
| Drivers and machine operators | 151 (0.18) | 0 | 136 (0.27) | 15 (0.23) | |
| Construction or mining workers | 55 (0.07) | 0 | 54 (0.11) | 1 (0.02) | |
| Package deliverers, cleaners, hand packers and other elementary workers | 444 (0.54) | 0 | 427 (0.86) | 17 (0.26) | |
| Full-time homemakers | 23,546 (28.4) | 23,546 (28.4) | 0 | 0 | |
| Students | 517 (0.62) | 517 (0.62) | 0 | 0 | |
| Unemployed | 2,945 (3.55) | 2,945 (3.55) | 0 | 0 | |
| Occupations not classified above | 1,388 (1.67) | 0 | 1,322 (2.67) | 66 (1.03) | |

**Table 4. Lifestyle attributes of the participants.**

| | All (N = 82,924) | Nonworkers (n = 27,008) | Workers without night shifts (n = 49,489) | Workers with night shifts (n = 6,427) | Missing value |
|---|---|---|---|---|---|
| **Smoking status, n (%)** | | | | | 546 |
| never_smoker | 48,272 (58.2) | 15,528 (57.5) | 29,019 (58.6) | 3,725 (58.0) | |
| Ex-smoker, stopped before learning of pregnancy | 19,210 (23.2) | 7,232 (26.8) | 10,674 (21.6) | 1,304 (20.3) | |
| Ex-smoker, stopped on awareness of pregnancy | 11,032 (13.3) | 2,866 (10.6) | 7,159 (14.5) | 1,007 (15.7) | |
| Current smoker | 3,864 (4.66) | 1,185 (4.39) | 2,315 (4.68) | 364 (5.66) | |
| **Alcohol use, n (%)** | | | | | 1,295 |
| Never drank | 27,218 (32.8) | 9,656 (35.8) | 15,623 (31.6) | 1,939 (30.2) | |
| Ex-drinker | 52,101 (62.8) | 16,165 (59.9) | 31,717 (64.1) | 4,219 (65.6) | |
| Current drinker | 2,310 (2.79) | 748 (2.77) | 1,381 (2.79) | 181 (2.82) | |
| **Mean caloric intake per day after awareness of pregnancy, (kcal (SD))[#]** | 1,732 (774) | 1,744 (726) | 1,722 (756) [‡] | 1,767 (1056) [§] | 667 |
| **Avoid gaining too much weight during pregnancy** | 77,892 (93.9) | 25,117 (93.0) | 46,675 (94.3) | 6,100 (94.9) | 950 |
| **presence of exercise habits in mid/late pregnancy** | | | | | |
| vigorous physical activity, (n (%)) | 2,687 (3.24) | 684 (2.53) | 1,600 (3.23) | 403 (6.27) | 1,329 |
| moderate physical activity, (n (%)) | 20,025 (24.1) | 6,879 (25.5) | 11,211 (22.7) | 1,935 (30.1) | 1,857 |
| walking for > 10 minutes consecutively on the short version of the International Physical Activity Questionnaire, (n (%)) | 58,004 (69.9) | 20,490 (75.9) | 32,973 (66.6) | 4,541 (70.7) | 2,427 |

#Mean caloric intake per day after awareness of pregnancy using the ANOVA and Bonferroni's multiple comparisons to examine which groups had significant differences.

§ $p < 0.05$ for the group of workers without night shift.

‡ $p < 0.05$ for the nonworker group.

the night shift group was significantly higher than those for the other groups. The aOR for the night shift group was also significant (nonworkers: 1.10, 95% CI 1.00–1.24; shift workers: 1.36, 95% CI 1.23–1.50; $p < 0.05$) (Fig 4). The cORs for consuming fast food more than three times a week in the last month of pregnancy for those who worked at least one night shift per month in early pregnancy were 0.91 (95% CI 0.77–1.08) for nonworkers and 1.48 (95% CI 1.15–1.88) for shift workers ($p < 0.05$) (Fig 5).but the aOR for the night shift group was not significantly higher than those for the other groups (nonworkers: 0.96, 95% CI 0.54–1.85; shift workers: 1.40, 95% CI 0.79–2.33; $p > 0.05$) (Fig 6).

**Table 5. Relationship between work status at the time of pregnancy awareness and unhealthy eating habits during pregnancy.**

| | All (N = 82,924) | Nonworker (n = 27,008) | Workers without shift work (n = 49,489) | Workers with shift work (n = 6,427) |
|---|---|---|---|---|
| **Eating fast food three or more times per week in mid/late pregnancy, n (%)** | 672 (0.81) | 198 (0.7) | 398 (0.8) | 76 (1.2) [§‡] |
| **Missing breakfast three or more times a week in a month of mid/late pregnancy, n (%)** | 4,695 (5.66) | 398 (0.8) | 1,456 (5.4) | 2,766 (5.6) |
| **Having a midnight snack three or more times per week in a month of mid/late pregnancy, n (%)** | 7,172 (8.65) | 2,211 (8.2) | 4,366 (8.8) [‡] | 595 (9.3) |
| **Eating snack foods once or more a week after becoming aware of pregnancy, n (%)** | 34,576 (41.7) | 11,415 (42.3) | 20,093 (40.6) [‡] | 3,068 (47.7) [§‡] |

Chi-square test was used to test for differences between the groups. Multiple comparisons using the Bonferroni method were performed.

§ $p < 0.05$ for the group of workers without night shift.

‡ $p < 0.05$ for the nonworker group.

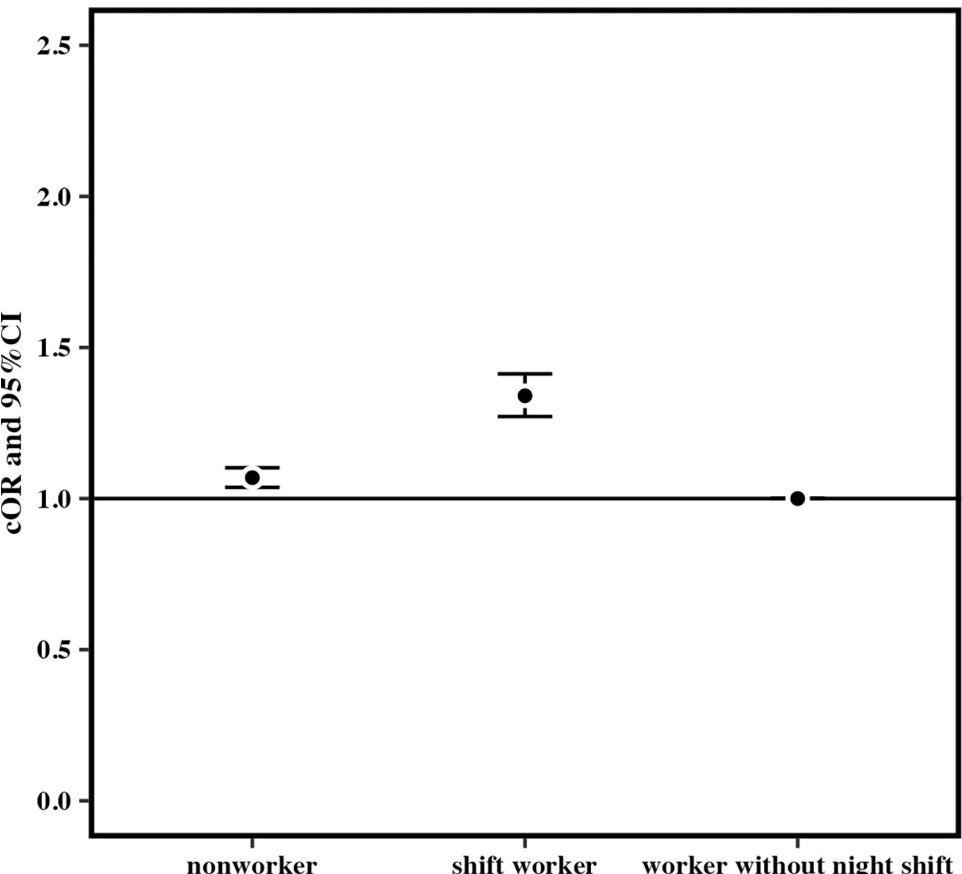

**Fig 3. The unadjusted odds ratio for eating snack foods more than once a week after becoming aware of pregnancy.**

When the reference group was workers without night shift, the cORs for mothers who had consumed snacks at least once a week after becoming aware of pregnancy for those who worked at least one night shift per month in early pregnancy were 1.06 (95% CI 1.03–1.10; p < 0.05) for nonworkers and 1.34 (95% CI 1.27–1.41) for shift workers (Fig 3). The cOR for the night shift group was significantly higher than those for the other groups. The aOR for the night shift group was also significant (nonworkers: 1.11, 95% CI 1.00–1.24; p > 0.05, shift workers: 1.36, 95% CI 1.23–1.50; p < 0.001) (Fig 4). The cORs for consuming fast food more than three times a week in the mid/late pregnancy for those who worked one and more night shifts per month in early pregnancy were 0.91 (95% CI 0.77–1.08) for nonworkers and 1.48 (95% CI 1.15–1.88) for shift workers (p < 0.01) (Fig 5). Nevertheless, the aOR for the night shift group was not significantly higher than those for the other groups (nonworkers: 0.97, 95% CI 0.54–1.85; shift workers: 1.40, 95% CI 0.79–2.33; p > 0.05) (Fig 6).

The present study carried out logistic regression and established the covariates as following: maternal age at gestation, maternal education, partner's education, presence of partner, presence of children at the time of pregnancy, annual income, existence of obesity before pregnancy, major occupational category at awareness of pregnancy, whether or not the participant was a medical professional (e.g., medical doctors, dentists, pharmacists, public health nurses, midwives, nurses, medical technicians, and other health workers), 6-item Kessler Psychological Distress Scale score in early and mid/late pregnancy (cut-off: 9 points), presence of

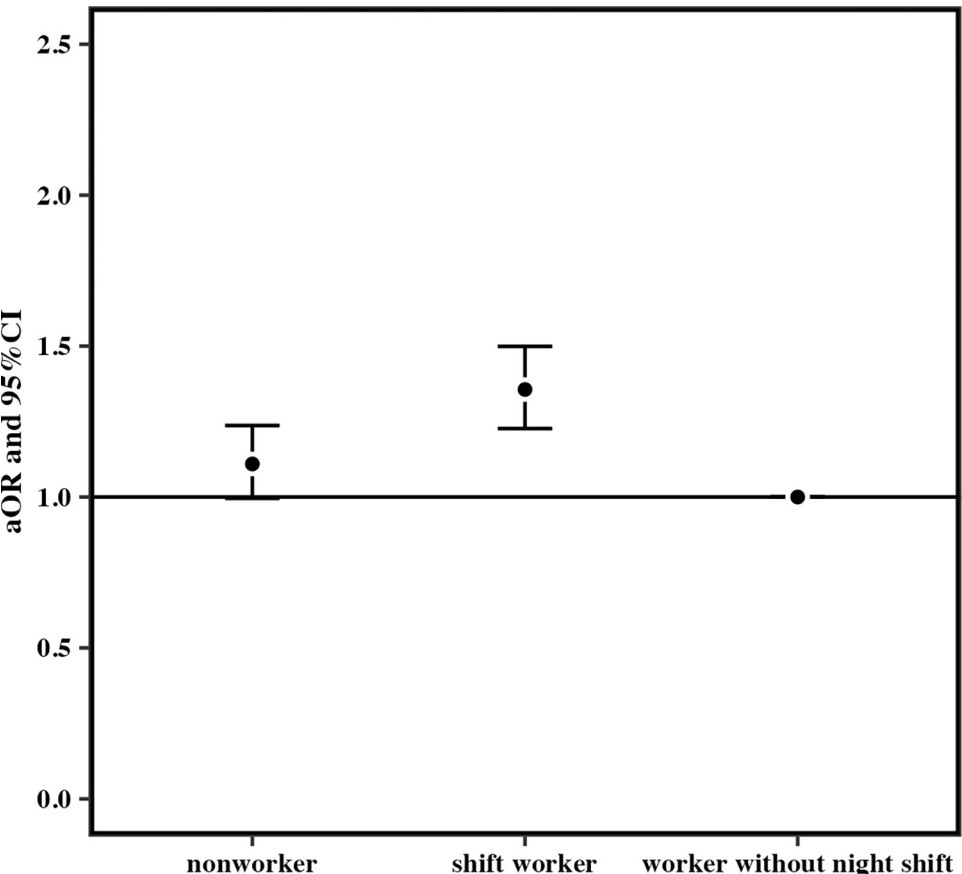

**Fig 4. Adjusted odds ratio for eating snack foods more than once a week after becoming aware of the pregnancy.**

maternal social support (company of friends/neighbors to whom you can talk casually about the participant's concern), presence of fertility treatment, history of miscarriage, history of alcohol consumption, history of smoking, presence of exercise habits (vigorous physical activity, moderate physical activity, and walking for > 10 minutes consecutively on the short version of the International Physical Activity Questionnaire) in mid/late pregnancy, history of morning sickness during the first 12 weeks of pregnancy, and whether the participant was aware of the need to avoid being overweight during pregnancy.

Working more than one night shift per month in early pregnancy increased the cOR and the aOR for habitual consumption of snack foods. There was also an association with the cOR for regular fast food consumption during pregnancy. However, the aOR was not insignificant.

As sensitivity analyses, we modified the definition for shift work and the cut-off values for habitual consumption of fast food, snacks, and midnight snacks. Changing these definitions did not significantly change the results. Similarly, there was no significant change in the results for the multiple substitution method.

## Discussion

Habitual consumption of snacks during pregnancy increased if the woman worked one and more night shifts per month in early pregnancy. Similarly, the unadjusted OR for regular fast food consumption during pregnancy also increased. The present study found that regular

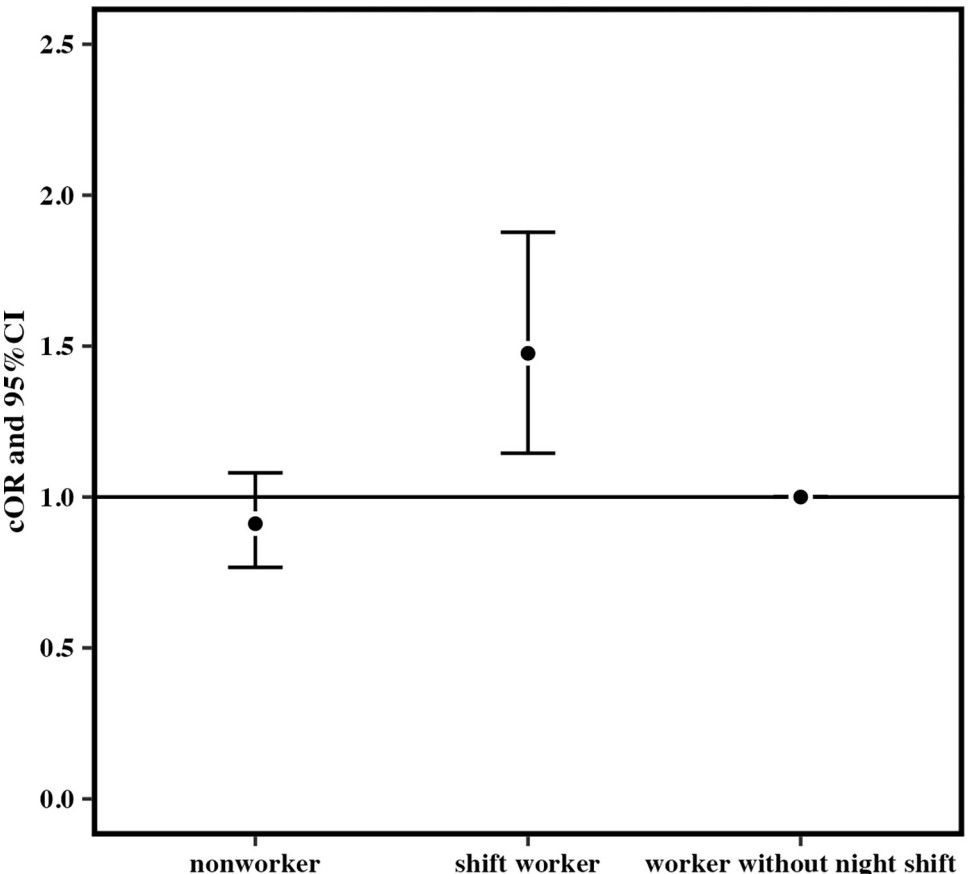

**Fig 5. The unadjusted odds ratio for eating fast food three or more times per week in mid/late pregnancy.**

night shifts (more than once a month) in early pregnancy may increase the likelihood of unhealthy eating habits, such as poor-quality snacking and increased fast food consumption.

This study found no significant association between night shift work and inappropriate weight gain during pregnancy compared to pregnant women who worked only day shifts. However, when verified against the new standards announced by the Japanese Society of Obstetrics and Gynecology in 2021, approximately 40% of pregnant women had too little weight gain during pregnancy based on the current standards in the present research [30]. This finding may have been associated with strict weight guidance in old standards [21, 29, 31]. In addition, pregnant women who could work rotating shifts during pregnancy may have been healthier than the other groups (healthy worker effect) [32]. Another possible effect may be the recent improvements in the working environment for pregnant women in Japan [33]. Factors other than night shift work that influence inappropriate weight gain during pregnancy require exploration in further studies.

The current study may be applicable when examining the relationship between shift work, improper eating habits, and poor nutrition during pregnancy. The present results may also help analyze the impact of shift work on problematic eating habits, regardless of pregnancy status.

The current study had several significant strengths. First, we explored night shifts and their effects on pregnant women. Several previous studies examined night shifts and obesity, as well as night shifts and unhealthy eating habits in workers [8–11]. However, most studies did not

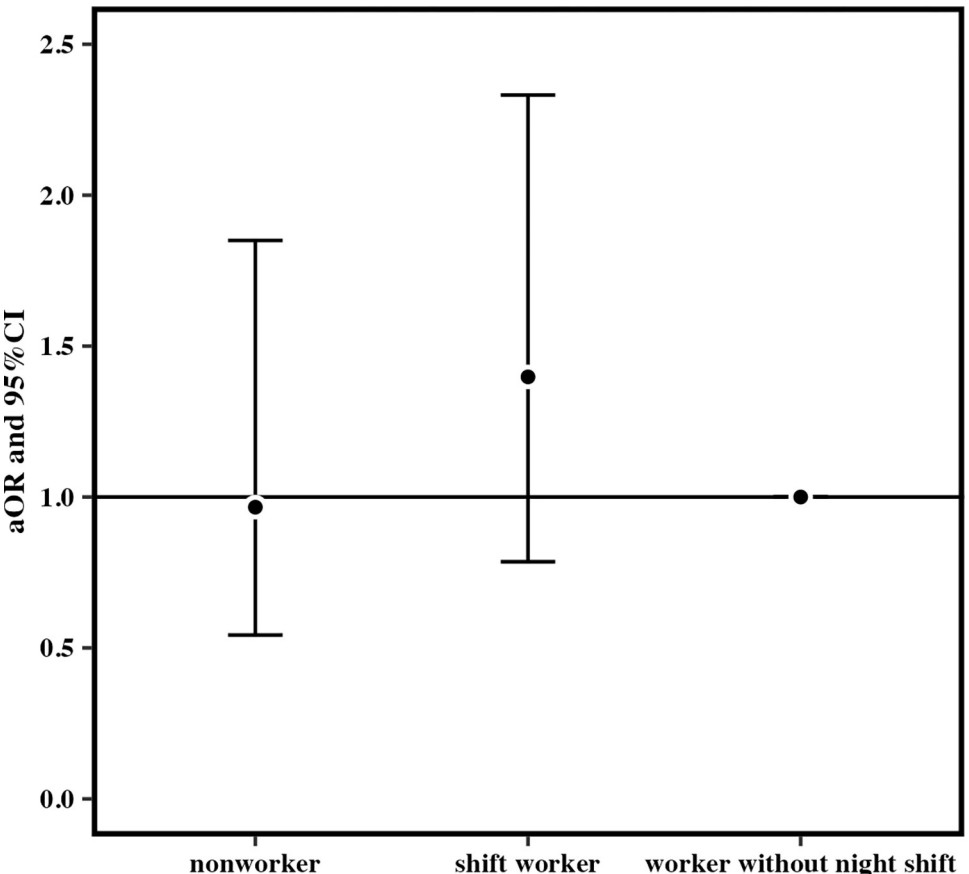

**Fig 6. Adjusted odds ratio for eating fast food three or more times per week in mid/late pregnancy.**

focus on pregnant women as the primary research target. The present study was significant in that it showed that night shifts may impact unhealthy eating habits among pregnant women.

Second, this study used data from a large-scale prospective cohort study. The JECS is a birth cohort survey covering all of Japan, and the overall sample size is approximately 100,000 people. Because most previous studies had small sample sizes (up to a few thousand) or used cross-sectional designs rather than cohort surveys, this study's large-scale cohort survey data represents a significant strength.

The present research involved several limitations. In this study, we compared the definition of night shift was limited to whether during the first trimester of pregnancy. Under the Japanese Labor Standards Law, pregnant women engaged in many types of labor in Japan take leave before childbirth, convert to day shift work only, or work shorter hours [33]. However, the dataset used in this study did not clarify until what point during pregnancy participants were engaged in night shifts or other work. This study's exact unemployment and work status duration was unknown because the JECS's survey was conducted based on tabulations responses to the survey instrument.

Moreover, shift workers, even those who worked shifts on one day per month, represented only approximately 10% of the participants in this study. Defining shift work in more narrow intervals may have resulted in insufficient eligible participants. Therefore, the present study could only examine the impacts of working up to one shift work per week in detail, and we could not have suggested a clear dose-response relationship between the number of night shifts

and the OR for snacking frequency. Future studies could conduct a more accurate evaluation of work status using timecard records and other data on working pregnant women. Regarding occupations involving shift work, this study could not examine participants by career in detail other than medical personnel. This study examined the frequency of consumption of snacks, which was the closest item to an assessment of potato chip consumption, which was implicated in previous studies as being related to night shifts. The results may contain some errors because examining the type of snack consumption in detail was impossible. There may also be a "healthy worker effect" or survivor bias if those who work night shifts are generally healthier than those who do not [32]. It is also possible that JECS participants are more health conscious than the general public, resulting in bias. These factors may have caused an underestimation of the effect of shift work in early pregnancy on unhealthy eating habits and inappropriate weight gain. In addition, other unknown covariates could not be examined in this study.

The present study suggested that shift work in early gestation may be associated with unhealthy dietary habits. More detailed future studies will be needed to clarify the impact of increased snack food consumption on infants' growth and childbirth.

## Conclusions

In conclusion, pregnant women who work one or more shifts per month in early pregnancy appear to have an increased frequency of consumption of fast food and snack foods compared with pregnant women who only work day shifts. Guidance on snack content and eating habits may be helpful for pregnant women working night shifts.

## Supporting information

**S1 Checklist. STROBE statement—checklist of items that should be included in reports of observational studies.**
(DOCX)

## Acknowledgments

We are grateful to all the participants in the study. Members of the JECS Group as of 2023: Michihiro Kamijima (principal investigator, Nagoya City University, Nagoya, Japan), Shin Yamazaki (National Institute for Environmental Studies, Tsukuba, Japan), Yukihiro Ohya (National Center for Child Health and Development, Tokyo, Japan), Reiko Kishi (Hokkaido University, Sapporo, Japan), Nobuo Yaegashi (Tohoku University, Sendai, Japan), Koichi Hashimoto (Fukushima Medical University, Fukushima, Japan), Chisato Mori (Chiba University, Chiba, Japan), Shuichi Ito(Yokohama City University, Yokohama, Japan), Zentaro Yamagata (University of Yamanashi, Chuo, Japan), Hidekuni Inadera (University of Toyama, Toyama, Japan), Takeo Nakayama(Kyoto University, Kyoto, Japan), Tomotaka Sobue (Osaka University, Suita, Japan),Masayuki Shima (Hyogo Medical University, Nishinomiya, Japan), Seiji Kageyama (Tottor iUniversity, Yonago, Japan), Narufumi Suganuma (Kochi University, Nankoku, Japan),Shoichi Ohga (Kyushu University, Fukuoka, Japan), and Takahiko Katoh (Kumamoto University, Kumamoto, Japan).

We thank Audrey Holmes, MA, and Benjamin Knight, MSc., from Edanz (https://jp.edanz.com/ac) for editing a draft of this manuscript.

## Author Contributions

**Conceptualization:** Satomi Funaki-Ishizu, Youichi Kurozawa.

**Formal analysis:** Satomi Funaki-Ishizu.

**Investigation:** Satomi Funaki-Ishizu.

**Methodology:** Toshio Masumoto, Shinji Otani, Youichi Kurozawa.

**Writing – original draft:** Satomi Funaki-Ishizu.

**Writing – review & editing:** Toshio Masumoto, Hiroki Amano, Shinji Otani, Youichi Kurozawa.

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
