## [Decision Letter · Decision Letter 0]

18 Jul 2023

PONE-D-23-09381Association between shift work in early pregnancy, snacking, and inappropriate weight gain during pregnancy: The Japan Environment and Children’s StudyPLOS ONE

Dear Dr. Funaki-Ishizu,

Thank you for submitting your manuscript to PLOS ONE. After careful consideration, we feel that it has merit but does not fully meet PLOS ONE’s publication criteria as it currently stands. Therefore, we invite you to submit a revised version of the manuscript that addresses the points raised during the review process.

We look forward to receiving your revised manuscript.

Kind regards,

Yosuke Yamada

Academic Editor

PLOS ONE

Journal Requirements:

“no”

“no”

“This study was funded by the Ministry of the Environment, Japan. The findings and conclusions of this article are solely the responsibility of the authors and do not represent the official views of the above government. We thank Audrey Holmes, MA, and Benjamin Knight, MSc., from Edanz (https://jp.edanz.com/ac) for editing a draft of this manuscript.”

“NO”

Reviewers' comments:

Reviewer's Responses to Questions

**Comments to the Author**

1. Is the manuscript technically sound, and do the data support the conclusions?

Reviewer #1: Yes

Reviewer #2: Yes

2. Has the statistical analysis been performed appropriately and rigorously? 

Reviewer #1: Yes

Reviewer #2: Yes

3. Have the authors made all data underlying the findings in their manuscript fully available?

Reviewer #1: No

Reviewer #2: No

4. Is the manuscript presented in an intelligible fashion and written in standard English?

Reviewer #1: Yes

Reviewer #2: Yes

5. Review Comments to the Author

Reviewer #1: This paper thoroughly examined the relationship between night shift work in early pregnancy and weight gain as well as snacking behavior during pregnancy, using large-scale data sets. The manuscript is well-organized and clearly presents the results, which provide valuable information for understanding the dietary challenges faced by pregnant women working night shifts. I think the sensitivity analysis has been carefully performed and the limitations have been well considered. However, one aspect that requires clarification is the analysis of the association between night shift work in early pregnancy and pre-pregnancy snacking behavior. The significance and interpretation of this relationship were somewhat unclear.

Major Concerns

The authors state their objective as "investigating the effects of night shifts on snacking behaviors during pregnancy," indicating their interest in exploring causal relationships. However, the analysis examines the relationship between night shifts in early pregnancy and the frequency of fast food consumption in the 20s prior to pregnancy, as well as the frequency of snack consumption such as potato chips in the year before pregnancy. Given the strict temporal ordering, this analysis appears to be inconsistent with the objective of investigating the 'effects' on snacking behaviors during pregnancy, as pre-pregnancy eating habits precede night work in early pregnancy. If the authors cannot provide justification for this analysis, it would be advisable to consider excluding the results pertaining to snacking behaviors before pregnancy from Table 3. Alternatively, the frequency of fast food and snack intake before pregnancy could be positioned as a baseline and used as an adjustment factor when examining the effects of night shifts in early pregnancy on snacking behavior in mid-to-late pregnancy.

Minor

Abstract & L27: 'evening meal'

Is the term 'evening meal' synonymous with 'midnight snack'? Please ensure consistent terminology.

L156: 'skipped breakfast less than three times a week were defined as habitually missing breakfast.'

I believe 'less than' should be replaced with 'more than' or 'at least.'

L163:

In this paper, the emphasis is solely on potato chips for snacking behavior. It was unclear whether the questionnaire included other types of snacks apart from potato chips. Providing more detailed explanations in the methods section would help clarify this aspect.

Table 2:

Some variable names begin with '-ing,' while others begin with 'Frequency.' It would be preferable to standardize the wording throughout the table, unless there is a specific intention behind this variation.

Reviewer #2: General comment

The work by Dr. Funakai and colleagues addresses an important issue: the association between shift work and inappropriate weight gain during pregnancy. Specifically, longitudinal data are used to examine the association between shift work in early pregnancy, snacking, and inappropriate weight gain during pregnancy. The authors conclude that pregnant women who work one or more shifts per month during early pregnancy consume fast food, potato chips, and other snack foods more frequently than pregnant women who work only day shifts. Of note, changing the definition of shift work and the definition of the cut-off points for habitual consumption of fast food, snacks, and late-night snacks did not significantly change the results.

Major comments

1. Energy intake is one of the most important factors in weight gain. However, this study appears to have conducted a food frequency survey but did not take energy intake into account when analyzing the data. Energy intake should be taken into account.

2. The authors note that there is no clear dose-response relationship between night work and metabolic diseases such as diabetes. Conversely, a previous study reported a dose-response relationship between the number of night shifts per month and the risk of diabetes (Céline Vetter et al., Diabetes Care, 41(4):762-769, 2018). In the present study, no clear association was found between night shifts and their frequency during pregnancy and weight gain. For the reader's understanding, this finding should be discussed in more detail. Please provide a citation.

3. The authors state in the Introduction that there is no evidence that shift work causes inappropriate weight gain or unhealthy snacking during pregnancy (lines 62-64). However, the authors discuss a previous study in the Discussion section of the manuscript (lines 282-284). The manuscript should be revised in the Introduction due to this inconsistency.

Minor comments

1. The tables are generally difficult to read and should be corrected. In particular, the tables need to be modified so that they fit into the manuscript.

2. In Table 2, there is a missing number in “Pregnant women who gained too little weight during pregnancy”.

3. Is the percentage of "Pregnant women whose weight gain was in the inappropriate range during pregnancy" the combined value of the following results? There are some incorrect values (Table 2).

4. Should not the physical activity levels be included in the results along with the other covariates?

5. Please provide a citation (lines 282-284).

6. PLOS authors have the option to publish the peer review history of their article (what does this mean?). If published, this will include your full peer review and any attached files.

Reviewer #1: No

Reviewer #2: No

---

## [Author Response · Author response to Decision Letter 0]

22 Aug 2023

Journal Requirements:

[Response]

As suggested, we have changed the manuscript format and file naming the title to PLOS ONE's style requirements.

“no”

[Response]

We appreciate your comments and the opportunity to clarify this critical point. This study was funded by the Ministry of Environment, Japan.

[Response]

We thank the editor for the positive comment. The funder had no role in the design and conduct of the study; collection, management, analysis, and interpretation of the data; preparation, review, or approval of the manuscript; and decision to submit the manuscript for publication.

[Response]

No. The authors received no salary from funders.

[Response]

The authors received no specific funding for this work.

“no”

[Response]

The authors have declared that no competing interests exist.

“This study was funded by the Ministry of the Environment, Japan. The findings and conclusions of this article are solely the responsibility of the authors and do not represent the official views of the above government. We thank Audrey Holmes, MA, and Benjamin Knight, MSc., from Edanz (https://jp.edanz.com/ac) for editing a draft of this manuscript.”

“NO”

[Response]

We thank the editor for the careful review. As suggested, we have removed the Funding Statement section and updated our Funding Statement.

[Response]

We appreciate the editor for the careful review. As suggested, we have removed the ethics statement from other sections besides the Methods.

5. Review Comments to the Author

Reviewer #1: 

This paper thoroughly examined the relationship between night shift work in early pregnancy and weight gain as well as snacking behavior during pregnancy, using large-scale data sets. The manuscript is well-organized and clearly presents the results, which provide valuable information for understanding the dietary challenges faced by pregnant women working night shifts. I think the sensitivity analysis has been carefully performed and the limitations have been well considered. However, one aspect that requires clarification is the analysis of the association between night shift work in early pregnancy and pre-pregnancy snacking behavior. The significance and interpretation of this relationship were somewhat unclear.

Major Concerns

1. The authors state their objective as "investigating the effects of night shifts on snacking behaviors during pregnancy," indicating their interest in exploring causal relationships. However, the analysis examines the relationship between night shifts in early pregnancy and the frequency of fast food consumption in the 20s prior to pregnancy, as well as the frequency of snack consumption such as potato chips in the year before pregnancy. Given the strict temporal ordering, this analysis appears to be inconsistent with the objective of investigating the 'effects' on snacking behaviors during pregnancy, as pre-pregnancy eating habits precede night work in early pregnancy. If the authors cannot provide justification for this analysis, it would be advisable to consider excluding the results pertaining to snacking behaviors before pregnancy from Table 3. Alternatively, the frequency of fast food and snack intake before pregnancy could be positioned as a baseline and used as an adjustment factor when examining the effects of night shifts in early pregnancy on snacking behavior in mid-to-late pregnancy.

[Response]

We thank the reviewer for these excellent comments. We agree with the reviewer’s observation that "pre-pregnancy eating habits precede night work in early pregnancy" and have removed snacking behaviors before pregnancy from Table 3.

Minor

2. Abstract & L27: 'evening meal'

Is the term 'evening meal' synonymous with 'midnight snack'? Please ensure consistent terminology.

[Response]

We have carefully reviewed the manuscript and corrected typos.

3. L156: 'skipped breakfast less than three times a week were defined as habitually missing breakfast.'

I believe 'less than' should be replaced with 'more than' or 'at least.'

[Response]

As suggested, we have changed the expression to "We defined those who said they skipped breakfast more than thrice a week as habitually missing it."

L163:

4. In this paper, the emphasis is solely on potato chips for snacking behavior. It was unclear whether the questionnaire included other types of snacks apart from potato chips. Providing more detailed explanations in the methods section would help clarify this aspect.

[Response]

We thank the reviewer for the positive comment. As the reviewer suggested, we carefully checked the questionnaire. The questionnaire 'Snack foods' included other snack foods besides potato chips. We added information about the definition of snack foods in the section "Frequency of consuming snack foods" (page 11, para 3.). We also discussed this issue in the Limitation section(page 31, para 1.)

5. Table 2:

Some variable names begin with '-ing,' while others begin with 'Frequency.' It would be preferable to standardize the wording throughout the table, unless there is a specific intention behind this variation.

[Response]

We have carefully reviewed and standardized the variable wordings.

Reviewer #2: General comment

The work by Dr. Funaki and colleagues addresses an important issue: the association between shift work and inappropriate weight gain during pregnancy. Specifically, longitudinal data are used to examine the association between shift work in early pregnancy, snacking, and inappropriate weight gain during pregnancy. The authors conclude that pregnant women who work one or more shifts per month during early pregnancy consume fast food, potato chips, and other snack foods more frequently than pregnant women who work only day shifts. Of note, changing the definition of shift work and the definition of the cut-off points for habitual consumption of fast food, snacks, and late-night snacks did not significantly change the results.

Major comments

1. Energy intake is one of the most important factors in weight gain. However, this study appears to have conducted a food frequency survey but did not take energy intake into account when analyzing the data. Energy intake should be taken into account.

[Response]

We thank the reviewer for these excellent comments. As requested, we have added this information to the result section (page 20, para 2).

2. The authors note that there is no clear dose-response relationship between night work and metabolic diseases such as diabetes. Conversely, a previous study reported a dose-response relationship between the number of night shifts per month and the risk of diabetes (Céline Vetter et al., Diabetes Care, 41(4):762-769, 2018). In the present study, no clear association was found between night shifts and their frequency during pregnancy and weight gain. For the reader's understanding, this finding should be discussed in more detail. Please provide a citation.

[Response]

We appreciate the reviewer's careful review. As requested, we have added this citation to the Introduction section. (page 3, para 2).

3. The authors state in the Introduction that there is no evidence that shift work causes inappropriate weight gain or unhealthy snacking during pregnancy (lines 62-64). However, the authors discuss a previous study in the Discussion section of the manuscript (lines 282-284). The manuscript should be revised in the Introduction due to this inconsistency.

[Response]

We appreciate the opportunity to clarify this point. There is much previous research on shift work and dietary habits, obesity, and non-communicable diseases such as hypertension and diabetes. However, most studies did not focus on gestational women as primary participants. In addition to the discussion section, we clarified that our study found no significant association between night shift work and inappropriate weight gain in pregnancy and discussed this issue (page 27, para 2). 

Minor comments

1. The tables are generally difficult to read and should be corrected. In particular, the tables need to be modified so that they fit into the manuscript.

[Response]

We have organized and modified the new tables to fit the manuscript.

2. In Table 2, there is a missing number in “Pregnant women who gained too little weight during pregnancy”.

[Response]

We added the number of “Pregnant women who gained too little weight during pregnancy” to Table 1.

3. Is the percentage of "Pregnant women whose weight gain was in the inappropriate range during pregnancy" the combined value of the following results? There are some incorrect values (Table 2).

[Response]

We thank the reviewer for the careful review and corrected the value. 

4. Should not the physical activity levels be included in the results along with the other covariates?

[Response]

We thank the reviewer for the positive comment. We added the results, physical activity levels, with other covariates in Table 4.

5. Please provide a citation (lines 282-284).

[Response]

Please see reviewer #2’s major comment No.3 and our comment.

---

## [Decision Letter · Decision Letter 1]

1 Sep 2023

Association between shift work in early pregnancy, snacking, and inappropriate weight gain during pregnancy: The Japan Environment and Children’s Study

PONE-D-23-09381R1

Dear Dr. Funaki-Ishizu,

We’re pleased to inform you that your manuscript has been judged scientifically suitable for publication and will be formally accepted for publication once it meets all outstanding technical requirements.

Kind regards,

Yosuke Yamada

Academic Editor

PLOS ONE

Additional Editor Comments (optional):

Reviewers' comments:

Reviewer's Responses to Questions

**Comments to the Author**

1. If the authors have adequately addressed your comments raised in a previous round of review and you feel that this manuscript is now acceptable for publication, you may indicate that here to bypass the “Comments to the Author” section, enter your conflict of interest statement in the “Confidential to Editor” section, and submit your "Accept" recommendation.

Reviewer #1: All comments have been addressed

Reviewer #2: All comments have been addressed

2. Is the manuscript technically sound, and do the data support the conclusions?

Reviewer #1: Yes

Reviewer #2: Yes

3. Has the statistical analysis been performed appropriately and rigorously? 

Reviewer #1: Yes

Reviewer #2: Yes

4. Have the authors made all data underlying the findings in their manuscript fully available?

Reviewer #1: No

Reviewer #2: No

5. Is the manuscript presented in an intelligible fashion and written in standard English?

Reviewer #1: Yes

Reviewer #2: Yes

6. Review Comments to the Author

Reviewer #1: (No Response)

Reviewer #2: (No Response)

7. PLOS authors have the option to publish the peer review history of their article (what does this mean?). If published, this will include your full peer review and any attached files.

Reviewer #1: No

Reviewer #2: No

---

## [Editor Report · Acceptance letter]

4 Oct 2023

PONE-D-23-09381R1 

Association between shift work in early pregnancy, snacking, and inappropriate weight gain during pregnancy: The Japan Environment and Children’s Study 

Dear Dr. Funaki-Ishizu:

I'm pleased to inform you that your manuscript has been deemed suitable for publication in PLOS ONE. Congratulations! Your manuscript is now with our production department. 

Kind regards, 

on behalf of

Dr. Yosuke Yamada 

Academic Editor

PLOS ONE